# A Bayesian Hierarchical Model for 2-by-2 Tables with Structural Zeros

James Stamey [1,*] and Will Stamey [2]

1 Department of Statistical Science, Baylor University, Waco, TX 76798, USA
2 Mendoza College of Business, University of Notre Dame, South Bend, IN 46556, USA; wstamey@nd.edu
* Correspondence: james_stamey@baylor.edu

**Abstract:** Correlated binary data in $2 \times 2$ tables have been analyzed from both the frequentist and Bayesian perspectives, but a fully Bayesian hierarchical model has not yet been proposed. This is a commonly used model for correlated proportions when considering, for example, a diagnostic test performance where subjects with negative results are tested a second time. We consider a new hierarchical Bayesian model for the parameters resulting from a $2 \times 2$ table with a structural zero. We investigate the performance of the hierarchical model via simulation. We then illustrate the usefulness of the model by showing how a set of historical studies can be used to build a predictive distribution for a new study that can be used as a prior distribution for both the risk ratio and marginal probability of a positive test. We then show how the prior based on historical $2 \times 2$ tables can be used to power a future study that accounts for pre-experimental uncertainty. High-quality prior information can lead to better decision-making by improving precision in estimation and by providing realistic numbers to power studies.

**Keywords:** meta-analytic prior; structural-zero; Bayesian

## 1. Introduction

In some experiment designs with a binary response, a second measurement is taken for some subjects, but not all. For example, Toyota et al. (1999) [1] consider data on a screening test for tuberculosis where only those testing disease-free on a first test are given a second test. Johnson and May (1995) [2] consider a common problem in medical studies where infected individuals are given a treatment and tested for improvement. In this situation, only those patients where no improvement is observed advance to a second phase and are again checked for improvement. Data from both of these examples lead to $2 \times 2$ tables where one cell is fixed in advance to be 0. Table 1 shows a table of this form. In the table, the cell counts are denoted $n_{ij}$ to denote rows and columns with corresponding cell probability $p_{ij}$. Subscripts $i$ and $j$ represent meeting the passing condition (=1) or not (=2) in the first and second phases, respectively. The passing condition could be a positive or negative test result (or framed as a failure or a success), depending on the problem context. In the study by Johnson and May (1995) [2], it is those who test negative for improvement who are passed to the next phase and administered yet another treatment. In contrast, some experiments involve passing only those testing positive to the second phase. Regardless of the framing, $n_{11}$ represents the count of observations meeting the passing condition in both phases, $n_{12}$ represents those passing in the first phase but not meeting the passing condition in the second, and $n_{22}$ represents the count of individuals who do not pass in the first phase and thus do not pass the second phase by default.

For a single table, the typical quantities of interest are the risk difference and risk ratio, which are often parameterized in terms of the [1, 1] cell probability $p_{11}$ and the marginal probability $\tau = p_{11} + p_{12}$. The risk difference is defined to be $d = \tau - p_{11}/\tau$, that is, the difference between the probability of a negative response on the initial test and the

conditional probability of a negative response on the second test given the initial negative response. The risk ratio is $RR = p_{11}/\tau^2$, the ratio of the probability of passing the second phase *conditional* on passing the first phase with the probability of passing the first phase.

There has been considerable interest in various approaches to estimate and perform hypothesis tests about the risk difference and risk ratio for a single $2 \times 2$ table with a structural zero. An example from veterinary medicine originally by Agresti (2012) [3] and recently analyzed by Lu et al. (2022) [4] tests a sample of calves for a primary pneumonia infection and checked again for a secondary infection. Calves who tested negative for the primary infection cannot develop the secondary infection, resulting in a natural structural zero. Using this example as a motivation, Lu et al. (2022) [4] consider various confidence interval procedures for the risk difference. Wang et al. (2024) [5] consider an exact interval for the risk ratio and compare their new interval to both frequentist and Bayesian intervals. They apply their new interval to the two-step tuberculosis data where only negative subjects are tested twice.

**Table 1.** Data and parameters for $2 \times 2$ table with structural zero.

| | **Phase II** | | |
|---|---|---|---|
| **Phase I** | + | - | **Total** |
| + | $n_{11}, p_{11}$ | $n_{12}, p_{12}$ | $n_1 = n_{11} + n_{12}, \tau = p_{11} + p_{12}$ |
| - | - | $n_{22}, p_{22}$ | $n_2 = n_{22}, p_2 = p_{22}$ |
| Total | | | $n = n_1 + n_2, 1 = \tau + p_2$ |

Though considerable work has been completed on estimating the parameters of the model for a single population, less work has been completed when multiple sources of information are available. Johnson and May (1995) [2] provide a frequentist approach to combining multiple tables, and Tang and Jiang (2011) [6] extend the model for a frequentist test of equality of risk ratios across tables, but to date, no Bayesian approach has looked at multiple tables. A reasonable approach to combine $2 \times 2$ tables from multiple trials, sites, etc., is to employ a Bayesian hierarchical model (Gelman et al.) [7]. The Bayesian hierarchical model enables a straightforward way to account for heterogeneity between the included studies and provides an operational approach to incorporate prior information and estimate numerous quantities simultaneously. The pooling of information across studies can provide more efficient estimation of all the study level risk ratios or risk differences. Also, the hierarchical model can easily facilitate the ranking and selection of sites by the effectiveness of a treatment or risk of infection.

The hierarchical model we propose here is similar in form to a random effects meta-analysis. Several authors, primarily in pharmaceutical applications, have discussed methods for using historical data to derive priors for a new study by predicting the parameter values based on a meta-analysis of historical studies. Sutton et al. (2007) [8] consider a hybrid Bayesian-frequentist approach to power a future study using a meta-analysis of historical studies to help determine the effect size. These priors are sometimes referred to as meta-analytic predictive (MAP) priors. Neuenschwander et al. (2010) [9] provide an early example. Weber et al. (2021) [10] provide a user-friendly R package to build MAP priors based on historical studies for experiments with binomial, Poisson, and normally distributed outcomes. These priors can be used as informative priors for new studies but can also be used as part of a sample size determination procedure. Recent examples include Du and Wang (2016) [11] and Qi et al. (2023) [12]. Yang et al. (2016) [13] and Fan et al. (2024) [14] focus especially on the value of meta-analysis in the case where the focal event is rare, such as the adverse effects of proposed interventions.

In this paper, we propose a hierarchical model for a series of $2 \times 2$ tables with a structural zero. The main model we consider is a Bayesian version of Tang and Jiang (2006) [6] where interest is in the risk ratio. The borrowing of strength across the studies improves inferences about the individual study parameters while also allowing for a variety of interesting statistical inference procedures. We apply the meta-analytic predictive prior method of Neuenschwander et al. (2010) [9] to develop a simulation-based induced prior

for the parameters of a new study. We illustrate how this prior can be used as a prior for data analysis and to determine the sample size required for a future study that achieves the desired power or probability of a successful trial. We propose an alternative hierarchical model in Section 5 for the case where the risk difference is of interest.

## 2. Bayesian Model

In this section, we describe the hierarchical Bayesian model we will use for inference for combining 2 × 2 tables with structural zeros. Suppose we observe data from $K$ sites or studies yielding 2 × 2 tables in the form of Table 1. We assume the parameters of the tables are exchangeable. The hierarchical model we propose is the following. For each study, we have count vector $\mathbf{z}_i = (n_{11}, n_{12}, n_{22})$, and

$$z_i \sim \text{trinomial}(N_i, \mathbf{p}_i) \tag{1}$$

where $N_i$ is the sample size of the $i^{th}$ study and $\mathbf{p}_i = (p_{11}, p_{12}, p_{22})$. For a single study, this distribution has the likelihood

$$L(p_{11i}, p_{12i} \mid z_i) \propto p_{11i}^{n_{11i}} p_{12i}^{n_{12i}} (1 - p_{11i} - p_{12i})^{N_i - n_{11i} - n_{12i}} \tag{2}$$

We reparameterize model (1), similar to Tang and Jiang (2011) [6], in terms of the nuisance parameter $\tau_i$ and risk ratio $RR_i$. The risk ratio is defined on the positive real line and $\tau_i$ is confined between 0 and $\min(1, 1/RR_i)$. The resulting likelihood is

$$L(RR_i, \tau_i) \propto \prod_{i=1}^{K} (RR_i \tau_i^2)^{n_{11i}} (\tau_i - RR_i \tau_i^2)^{n_{12i}} (1 - \tau_i)^{N_i - n_{11i} - n_{12i}}. \tag{3}$$

In the most general case, we consider a hierarchical model on the log risk ratios and model the $\tau_i$'s with a hierarchical truncated beta distribution. For the risk ratios, we have

$$\Psi_i = log(RR_i) \sim \mathcal{N}(\mu_\Psi, \sigma_\Psi^2). \tag{4}$$

Modeling these study-specific parameters with a normal distribution with shared hyper-parameters allows for "borrowing" or "pooling" of strength when estimating each of the study-level effects and can be particularly beneficial when the sample sizes are small. In general, we would imagine little information would be available on the parameters $\mu_\Psi$ and $\sigma_\Psi$. Reasonable weakly informative priors for the means would be $\mu_\Psi \sim \mathcal{N}(0, \nu_\Psi^2)$, where $\nu_\Psi^2$ is a suitably chosen large number such as 100. Some controversy exists on what a suitable "non-informative" prior for the variance parameter is in hierarchical models such as the one we propose here. Historically, an inverse-gamma$(\epsilon, \epsilon)$ with $\epsilon$ often chosen to be 0.001 has commonly been used. Gelman (2013) [7] has shown that this parameterization has many weaknesses, especially in the case of a small number of strata. Gelman (2013) [7] determined that the uniform distribution, half normal, and half t distributions as a prior for the standard deviation all outperform the inverse gamma for the variance. We provide code for both the cases of a uniform prior or a half-normal prior for the standard deviation $\sigma_\Psi$. That is, either $\sigma \sim \text{Uniform}(0, B)$ or $\sigma \sim \mathcal{HN}(\sigma_0)$, where $B$ is the upper bound of the uniform prior, and $\sigma_0$ is the scale parameter of the half-normal.

The model for the $\tau$'s requires careful thought due to the restriction that they are bounded by the minimum of 1 and $1/RR_i$. One common hierarchical model for probabilities is to assume

$$\phi_i = \text{logit}(\tau) \sim \mathcal{N}(\mu_\phi, \sigma_\phi^2). \tag{5}$$

However, given the upper bound problem, we chose instead to remain on the probability scale. For this version of the hierarchical model, conditional on $RR_i$, the $\tau$'s are assumed to have a truncated beta distribution,

$$\tau | RR_i \sim \text{beta}(a, b) I(0, \mathbf{min}(1, 1/RR_i)) \tag{6}$$

We reparameterize the beta in terms of $\mu_\tau = \frac{a}{a+b}$ and $\rho_\tau = a + b$. The grand mean, $\mu_\tau$, is assigned a beta$(c, d)$ prior, and $\rho_\tau$ is given a gamma$(e, f)$ prior. The parameters of these distributions are generally selected to be weakly informative. The joint posterior distribution of all the parameters is the product of the likelihood in (3), the normal distributions for the log risk ratio in (4), the beta distributions for the $\tau_i$'s, and the priors for the parameters at the top of the hierarchy. There is no apparent closed form for any of the parameters of this model. We use the software JAGS version 4.3.2 to perform inference. The code is available at https://github.com/will-stamey1/metaanalytic_for_2by2s. An alternative formulation of the hierarchical model that is more convenient for some scenarios is provided in Section 5.

*Meta Analytic-Based Prior*

The hierarchical model described above can be used to perform inferences on the parameters of interest but can also be used to determine a prior for a new study. Our framework is similar to that of Sutton et al. (2007) [8], Neuenschwander et al. (2010) [9], and Qi et al. (2023) [12]. As mentioned previously, we assume the parameters in the different studies to be exchangeable. This calls for careful selection of the included studies.

To determine the meta-analytic prior, we augment the likelihood and priors of the historical studies described above with a new study with yet-to-be-observed data treated as missing. This new study with parameters $RR^*$ and $\tau^*$ is incorporated into the MCMC scheme. The model, based on the borrowed information from the historical studies, provides what are essentially prior-predictive distributions for both the study-level parameters and the unobserved data of the new study.

In order to obtain the predictive distribution for the parameters of the new study, we add the following trinomial distribution to the likelihood of the hierarchical model:

$$z^* \sim \text{trinomial}(N^*, \mathbf{p}^*) \tag{7}$$

where $p^*$ relates to parameters $\tau^*$ and $RR^*$ following the reparameterization of the hierarchical model before.

The predictive distribution for the parameters can be used as a prior distribution for the parameters of the new study. Weber et al. [10] use a mixture of parametric distributions to approximate the prior with the R package RBesT. The Monte Carlo samples from the meta-analysis can also be used as a numerical approximation to the prior. The prior predictive distribution of the data for the new study can be used to simulate likely values for the data in order to either power a new study or look at the probability of a successful trial for a specific sample size.

## 3. Simulation Study

We conducted a simulation to investigate both the performance of the hierarchical model and the impact of between-trial variability on the informativeness of the resulting prior. We expand on the number of groups but use the data of Tang and Jiang (2011) [6] to determine the parameter values for our first simulation. The dataset they examine is for a two-phase treatment regimen in which a sample of patients first receives an initial treatment. Those who do not show improvement then proceed to a follow-up treatment. The parameter of interest is the risk ratio. Values less than 1 indicate that the second regimen leads to an improvement. In the work of Tang and Jiang [6], the groups are based on the severity of disease. This would likely violate the exchangeabilty assumption of our model, but the example works as an illustration. They find the $\tau$s' center approximately around 0.4 and the risk ratios' center approximately at 0.7. Therefore, for our simulation, we chose $\mu_\tau = 0.4$, $\rho_\tau = 20$, and $\mu_\Psi = \log(0.7)$. We also looked at the case where the risk ratio is centered above one to make sure performance was not impacted in that case.

We are interested in the impact of the between-study variability of the risk ratios, so we consider two values of $\sigma_\Psi$, 0.075 and 0.15. Also of interest is the effect of the amount of information. To observe this, we vary the number of studies (5, 25) and the number of observations in each study. The sample sizes for each study are randomly generated from either (50, 100) or (100, 200). For these 16 combinations, we generated 1000 datasets for each combination. For each dataset, the values at the top of the hierarchy remain the same, but, the study level $\tau_i$'s and $RR_i$'s vary. Therefore, for the simulation, we keep track of the average posterior mean, standard deviation, and coverage of 95% intervals for $\mu_\phi$, $\sigma_\phi$, $\mu_\tau$, and $\rho_\tau$. For the study-level parameters, we keep track of the total average bias for all the $\tau_i$'s and $RR_i$ and the overall coverage of the 95% intervals. The total average bias is calculated as

$$\text{Average Percent Bias} = \frac{1}{B \cdot K} \sum_1^B \sum_{i=1}^K \left( \frac{\hat{\tau}_i - \tau_i}{\tau_i} \right) \times 100 \tag{8}$$

and

$$\text{Average Percent Bias} = \frac{1}{B \cdot K} \sum_1^B \sum_{i=1}^K \left( \frac{\hat{RR}_i - RR_i}{RR_i} \right) \times 100 \tag{9}$$

where $\hat{\tau}_{1i}$ is the posterior mean of $\tau_i$, and $\hat{RR}_i$ is the posterior mean for $RR_i$.

The models converged successfully, with simulation runs consistently showing Gelman–Rubin statistics equal too or extremely close to 1 for all parameters. In Table 2, we provide the bias, confidence interval width, and coverage of 95% intervals for the parameters at the highest level of the risk ratio. In all cases, the coverage is at or above nominal, and the intervals become more narrow as both the sample size and number of studies increase. The estimate for the between-trial standard deviation exhibits substantial bias for most parameter and sample size combinations. We considered both the half-normal and uniform priors for the standard deviation, and in both cases, the posterior mean was significantly higher than the true value. However, the coverage was still nominal and the bias did lessen as the number of studies increased.

**Table 2.** Simulation results for $\mu_{ln(RR)}$ and $\sigma_{ln(RR)}$.

| | | | | $\mu_{\textbf{ln(RR)}}$ | | | $\sigma_{\textbf{ln(RR)}}$ | | |
|---|---|---|---|---|---|---|---|---|---|
| $\sigma$ | K | N | RR | % Bias | CI Len. | Covrg. | % Bias | CI Len. | Covrg. |
| 0.075 | 5 | ∼**unif**(100, 200) | 0.7 | 6.779 | 0.751 | 0.997 | 245.446 | 0.745 | 0.976 |
| 0.075 | 5 | ∼**unif**(50, 100) | 0.7 | 13.136 | 0.986 | 0.994 | 336.613 | 0.841 | 0.968 |
| 0.075 | 25 | ∼**unif**(100, 200) | 0.7 | 1.85 | 0.209 | 0.961 | 42.579 | 0.242 | 0.98 |
| 0.075 | 25 | ∼**unif**(50, 100) | 0.7 | 3.873 | 0.3 | 0.977 | 99.01 | 0.351 | 0.969 |
| 0.15 | 5 | ∼**unif**(100, 200) | 0.7 | 6.082 | 0.806 | 0.99 | 95.032 | 0.766 | 0.963 |
| 0.15 | 5 | ∼**unif**(50, 100) | 0.7 | 14.55 | 1.014 | 0.988 | 131.355 | 0.849 | 0.988 |
| 0.15 | 25 | ∼**unif**(100, 200) | 0.7 | 0.866 | 0.237 | 0.954 | 5.859 | 0.284 | 0.97 |
| 0.15 | 25 | ∼**unif**(50, 100) | 0.7 | 2.465 | 0.317 | 0.957 | 21.12 | 0.384 | 0.975 |
| 0.075 | 5 | ∼**unif**(100, 200) | 1.2 | 2.147 | 0.693 | 0.997 | 217.951 | 0.706 | 0.977 |
| 0.075 | 5 | ∼**unif**(50, 100) | 1.2 | −1.746 | 0.918 | 0.995 | 306.185 | 0.813 | 0.971 |
| 0.075 | 25 | ∼**unif**(100, 200) | 1.2 | 2.047 | 0.192 | 0.968 | 30.963 | 0.218 | 0.982 |
| 0.075 | 25 | ∼**unif**(50, 100) | 1.2 | 5.965 | 0.271 | 0.974 | 74.625 | 0.302 | 0.978 |
| 0.15 | 5 | ∼**unif**(100, 200) | 1.2 | 1.285 | 0.761 | 0.994 | 85.275 | 0.741 | 0.97 |
| 0.15 | 5 | ∼**unif**(50, 100) | 1.2 | −5.67 | 0.961 | 0.994 | 121.687 | 0.825 | 0.965 |
| 0.15 | 25 | ∼**unif**(100, 200) | 1.2 | −0.573 | 0.228 | 0.953 | 6.104 | 0.265 | 0.968 |
| 0.15 | 25 | ∼**unif**(50, 100) | 1.2 | 1.593 | 0.297 | 0.958 | 15.18 | 0.345 | 0.98 |

Table 3 provides the same results for the model for the marginal probability. Again, coverage for both of the parameters was at or above nominal, and the bias decreased as the sample size and number of studies increased.

Table 4 provides results for the study-level parameters. These parameters changed with each dataset, which is why we looked at the overall bias using Equations (8) and (9) and the combined coverage. As can be seen, at the study level, the parameters are well estimated with only moderate biases in the case of only five studies and smaller sample sizes. The other combinations have relatively low bias and good coverage.

Finally, Table 5 provides the average of the predicted distributions for the parameters of the new studies. As can be seen, the predictive distributions are centered approximately at the mean of the population distribution, and the 95% interval widths decrease for the larger study and sample size cases demonstrating that the method would provide reasonable prior distributions based on these historical studies.

**Table 3.** Simulation results for $\mu_\tau$ and $\rho_\tau$.

| | | | | $\mu_p$ | | | $\rho_\tau$ | | |
|---|---|---|---|---|---|---|---|---|---|
| $\sigma$ | K | N | RR | % Bias | CI Len. | Covrg. | % Bias | CI Len. | Covrg. |
| 0.075 | 5 | ∼**unif**(100, 200) | 0.7 | 2.577 | 0.249 | 0.985 | −13.426 | 38.308 | 0.981 |
| 0.075 | 5 | ∼**unif**(50, 100) | 0.7 | 3.873 | 0.262 | 0.989 | −15.567 | 38.684 | 0.982 |
| 0.075 | 25 | ∼**unif**(100, 200) | 0.7 | 0.472 | 0.095 | 0.959 | 0.23 | 23.548 | 0.966 |
| 0.075 | 25 | ∼**unif**(50, 100) | 0.7 | 0.242 | 0.099 | 0.973 | 2.852 | 26.574 | 0.97 |
| 0.15 | 5 | ∼**unif**(100, 200) | 0.7 | 2.746 | 0.251 | 0.991 | −15.013 | 37.598 | 0.969 |
| 0.15 | 5 | ∼**unif**(50, 100) | 0.7 | 4.193 | 0.259 | 0.985 | −13.742 | 39.473 | 0.989 |
| 0.15 | 25 | ∼**unif**(100, 200) | 0.7 | 0.689 | 0.095 | 0.954 | 0.157 | 23.599 | 0.965 |
| 0.15 | 25 | ∼**unif**(50, 100) | 0.7 | 0.881 | 0.1 | 0.961 | 1.495 | 26.311 | 0.972 |
| 0.075 | 5 | ∼**unif**(100, 200) | 1.2 | 4.216 | 0.262 | 0.992 | −13.78 | 39.039 | 0.981 |
| 0.075 | 5 | ∼**unif**(50, 100) | 1.2 | 5.477 | 0.281 | 0.986 | −15.86 | 39.649 | 0.994 |
| 0.075 | 25 | ∼**unif**(100, 200) | 1.2 | 1.042 | 0.103 | 0.948 | 0.83 | 24.897 | 0.965 |
| 0.075 | 25 | ∼**unif**(50, 100) | 1.2 | 2.055 | 0.116 | 0.957 | 2.36 | 28.298 | 0.979 |
| 0.15 | 5 | ∼**unif**(100, 200) | 1.2 | 3.641 | 0.264 | 0.99 | −15.294 | 38.536 | 0.982 |
| 0.15 | 5 | ∼**unif**(50, 100) | 1.2 | 5.686 | 0.282 | 0.984 | −16.821 | 39.31 | 0.989 |
| 0.15 | 25 | ∼**unif**(100, 200) | 1.2 | 0.554 | 0.104 | 0.963 | 0.268 | 25.368 | 0.973 |
| 0.15 | 25 | ∼**unif**(50, 100) | 1.2 | 1.257 | 0.117 | 0.965 | 1.031 | 28.325 | 0.971 |

**Table 4.** Simulation results for study-level parameters.

| | | | | $RR_i$ | | | $\tau$ | | |
|---|---|---|---|---|---|---|---|---|---|
| $\sigma$ | K | N | RR | % Bias | CI Len. | Covrg. | % Bias | CI Len. | Covrg. |
| 0.075 | 5 | ∼**unif**(100, 200) | 0.7 | 0.857 | 0.543 | 0.978 | 0.43 | 0.145 | 0.956 |
| 0.075 | 5 | ∼**unif**(50, 100) | 0.7 | 2.13 | 0.77 | 0.98 | 1.193 | 0.199 | 0.96 |
| 0.075 | 25 | ∼**unif**(100, 200) | 0.7 | 0.39 | 0.321 | 0.975 | 0.794 | 0.137 | 0.952 |
| 0.075 | 25 | ∼**unif**(50, 100) | 0.7 | 0.641 | 0.461 | 0.988 | 1.543 | 0.185 | 0.953 |
| 0.15 | 5 | ∼**unif**(100, 200) | 0.7 | 1.554 | 0.562 | 0.962 | 0.222 | 0.145 | 0.957 |
| 0.15 | 5 | ∼**unif**(50, 100) | 0.7 | 2.303 | 0.776 | 0.972 | 1.167 | 0.2 | 0.961 |
| 0.15 | 25 | ∼**unif**(100, 200) | 0.7 | 1.602 | 0.397 | 0.941 | 0.874 | 0.139 | 0.949 |
| 0.15 | 25 | ∼**unif**(50, 100) | 0.7 | 2.231 | 0.516 | 0.962 | 1.551 | 0.187 | 0.951 |
| 0.075 | 5 | ∼**unif**(100, 200) | 1.2 | 3.496 | 0.866 | 0.98 | 0.436 | 0.138 | 0.957 |
| 0.075 | 5 | ∼**unif**(50, 100) | 1.2 | 5.438 | 1.243 | 0.976 | 0.46 | 0.191 | 0.96 |
| 0.075 | 25 | ∼**unif**(100, 200) | 1.2 | 1.327 | 0.507 | 0.972 | 0.709 | 0.127 | 0.952 |
| 0.075 | 25 | ∼**unif**(50, 100) | 1.2 | 2.721 | 0.709 | 0.982 | 1.456 | 0.174 | 0.958 |
| 0.15 | 5 | ∼**unif**(100, 200) | 1.2 | 3.339 | 0.925 | 0.962 | 0.345 | 0.139 | 0.953 |
| 0.15 | 5 | ∼**unif**(50, 100) | 1.2 | 5.322 | 1.295 | 0.967 | 0.817 | 0.192 | 0.956 |
| 0.15 | 25 | ∼**unif**(100, 200) | 1.2 | 1.561 | 0.656 | 0.944 | 1.107 | 0.132 | 0.951 |
| 0.15 | 25 | ∼**unif**(50, 100) | 1.2 | 2.821 | 0.827 | 0.955 | 1.738 | 0.177 | 0.95 |

**Table 5.** Simulation results for new study parameters.

| | | | | $RR^*$ | | $\tau^*$ | |
|---|---|---|---|---|---|---|---|
| $\sigma$ | K | N | RR | Mean | CI Length | Mean | CI Length |
| 0.075 | 5 | $\sim$**unif**(100, 200) | 0.7 | 0.745 | 1.22 | 0.41 | 0.595 |
| 0.075 | 5 | $\sim$**unif**(100, 200) | 1.2 | 1.3 | 1.981 | 0.417 | 0.605 |
| 0.075 | 5 | $\sim$**unif**(50, 100) | 0.7 | 0.765 | 1.591 | 0.415 | 0.61 |
| 0.075 | 5 | $\sim$**unif**(50, 100) | 1.2 | 1.35 | 2.635 | 0.422 | 0.621 |
| 0.075 | 25 | $\sim$**unif**(100, 200) | 0.7 | 0.703 | 0.402 | 0.402 | 0.453 |
| 0.075 | 25 | $\sim$**unif**(100, 200) | 1.2 | 1.216 | 0.642 | 0.404 | 0.456 |
| 0.075 | 25 | $\sim$**unif**(50, 100) | 0.7 | 0.706 | 0.576 | 0.401 | 0.454 |
| 0.075 | 25 | $\sim$**unif**(50, 100) | 1.2 | 1.235 | 0.892 | 0.408 | 0.462 |
| 0.15 | 5 | $\sim$**unif**(100, 200) | 0.7 | 0.76 | 1.363 | 0.411 | 0.6 |
| 0.15 | 5 | $\sim$**unif**(100, 200) | 1.2 | 1.324 | 2.277 | 0.415 | 0.609 |
| 0.15 | 5 | $\sim$**unif**(50, 100) | 0.7 | 0.77 | 1.658 | 0.417 | 0.604 |
| 0.15 | 5 | $\sim$**unif**(50, 100) | 1.2 | 1.363 | 2.851 | 0.423 | 0.625 |
| 0.15 | 25 | $\sim$**unif**(100, 200) | 0.7 | 0.713 | 0.552 | 0.403 | 0.454 |
| 0.15 | 25 | $\sim$**unif**(100, 200) | 1.2 | 1.223 | 0.944 | 0.402 | 0.458 |
| 0.15 | 25 | $\sim$**unif**(50, 100) | 0.7 | 0.716 | 0.678 | 0.404 | 0.457 |
| 0.15 | 25 | $\sim$**unif**(50, 100) | 1.2 | 1.237 | 1.112 | 0.405 | 0.465 |

## 4. Sample Size Determination

We now illustrate how the prior distribution generated from the historical studies can be used to power a future study. Without loss of generality, we assume the hypothesis of interest is:

$$H_0 : RR^* = 1$$

$$H_a : RR^* < 1$$

The sample size determination procedure we consider here is simulation-based and follows from [15], which has been modified to use with MAP priors by, for example, Qi et al. (2023) [12]. The method allows for distinct priors for two different parts of the procedure. The design prior is necessarily informative and is used to predict what future data will look like. The design prior can be constructed from historical data or expert opinion. It is important to note that the design prior is similar to "design parameters" in frequentist sample size determination and should match study goals. For instance, for a frequentist sample size determination, the study is powered for a particular effect size of interest that matches, for instance, regulatory requirements or effects observed in previous studies. In contrast, the design prior specifies a distribution of that parameter on which the probability of detection is conditioned. This results in a probability of rejection/success, etc., given the parameter is distributed in the manner specified by the design prior.

At the design stage, design priors can account for uncertainty regarding both nuisance parameters and primary parameters, or just in the nuisance parameters. That is, a prior can be assigned to the nuisance parameter(s) to account for pre-experimental uncertainty while a fixed value is assigned to the parameter of interest in the manner of frequentist sample-size planning. The fixed value is usually set at some practically significant threshold. When a design prior is used for the focal parameter, the resulting rejection rate is often referred to as "assurance" or "Bayesian assurance" instead of power; see, for example, Pan and Bannerjee (2023) [16]. The algorithm we propose is provided below.

1. Using historical studies, fit the meta-analysis model and determine the design priors for the current study parameters.
2. Sample values from $RR^*$ and $\tau^*$. A different value for $RR^*$ is drawn if interest is in assurance and is fixed at a particular value for power.
3. For a sample of size n*, simulate new study data based on RR* and p*.
4. Analyze the data (with either non-informative priors or using the informative prior) and determine posterior probability $RR^* < 1$.

5.    Repeat steps 2–4 a large number of times and calculate power (or assurance for random RR\*) by computing the proportion of datasets where the null is rejected.
6.    Repeat steps 2–5 with different sample sizes to obtain the desired power/probability of a successful trial.

As an example, to formulate a design prior, we simulated 25 datasets of size $n = 30$ each. For this example, the global risk ratio was assumed to be 0.55, and the mean of the marginal probabilities was assumed to be 0.4. Suppose we seek a power of 0.8, Type I error probability of 0.05, and want to find the sample size to detect a risk ratio of 0.55. At the design stage, considering both the risk ratio and the marginal probability as fixed yields sample sizes similar to Tang et al. (2006) [17]. The fully Bayesian approach allows for the incorporation of uncertainty at the design stage. Figure 1 provides the powers/assurance for three cases. The first is where the risk ratio is assumed as fixed at 0.55 and the marginal probability at 0.366. The second replaces the fixed value of $\pi_{1+}$ with the induced prior from the predictive distribution based on the hierarchical model. Not surprisingly, this uncertainty at the design stage leads to a larger required sample size. Finally, we replaced the fixed value of the risk ratio with the predictive distribution to allow for uncertainty in both parameters. The resulting probability is no longer "power" because it is not a probability of rejection for a specific value. This quantity is commonly called assurance, and depending on the amount of variability in the design prior for the parameter of interest, the assurance might not converge to one as the sample size increases.

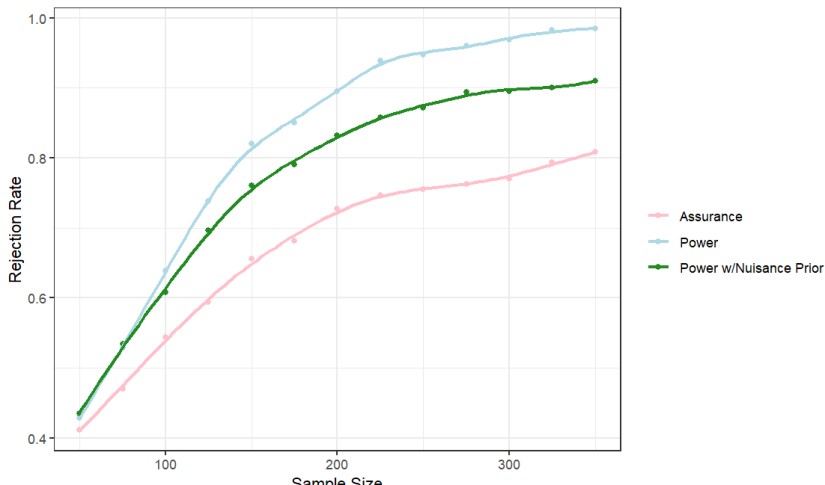

**Figure 1.** Rejection rates across sample sizes. The "Assurance" condition uses design priors on both $RR$ and $\tau$, while "Power" uses fixed values for each, and "Power w/Nuisance Prior" uses a fixed value for $RR$, the focal parameter, with a design prior on the nuisance parameter $\tau$.

To achieve a power/assurance of 0.8, we require approximately 140 observations if both parameters are assumed fixed at the design stage. If uncertainty in $\tau$ is accounted for at the design stage with the prior from the previous studies, then a sample size of approximately 180 observations is required. If the historical studies are used to provide informative priors for both parameters at the design stage, a sample of approximately 330 observations is required to obtain an assurance of 0.8.

## 5. Example Walkthrough

In this section, we walk through a single example illustrating how to estimate parameters and develop a prior for 2 × 2 tables with structural zeros. Suppose an investigator wants to conduct a new experiment investigating the relative effect of two treatments for a disease. The first treatment is fairly unobtrusive and can be taken at home while the second involves a more intensive and costly inpatient care regime at a clinic. Because of this

difference, the unobtrusive treatment is attempted first for each patient, and the inpatient care regime is only tested on the patients for whom the first treatment is deemed ineffective.

The investigator wants to estimate the risk ratio of these two treatments, i.e., the ratio of the probability of failure of the inpatient care regime (given the first treatment was unsuccessful) to the probability of failure of the first treatment. The investigator seeks to derive an informed prior for the new experiment from historic information. Eight prior studies have already been conducted in different cities and at different times. Since the nature of disease can vary based on location (due to economic differences) and time (due to shocks), the risk ratio is expected to vary across these experiments. However, the phenomenon being studied is the same, so we should expect that the risk ratios from the different experiments are related, and we would want to borrow information across the trials. A hierarchical model-based prior can be used to synthesize the historic information in a principled way.

We generate a dataset of 8 2 × 2 tables with sample sizes randomly generated between 200 and 300 participants. The true mean of the risk ratios is 0.5, indicating that the second treatment (administered after failure of the first) fails to improve the patients half as often as the first. The individual experiments' log risk ratios are generated around this mean with a standard deviation of 0.3 (we transform the risk ratios using the log in order to realistically use the normal distribution). We generate the parameter $\tau$ with a centrality parameter 0.4 and a value of $\rho$ of 20. We place relatively diffuse priors on the hierarchical parameters of the log risk ratio.

Using the R functions available on the GitHub page, we perform the analysis with the following function:

```
history <- make_history(
            # History Data: ###########
            data = counts_data
            # Prior Parameters: #######
            prior_lnRRmu_mu = log(1), prior_lnRRmu_tau = 1/100,
            prior_lnRRtau_lo = 0, prior_lnRRtau_hi = 1,
            prior_p1mu_alpha = 1, prior_p1mu_beta = 1,
            prior_p1rho_alpha = 1, prior_p1rho_beta = 1)
```

Here, `counts_data` holds a data frame of counts where each row is a single experiment and each column is the counts of a possible outcome, corresponding to the cells in Table 1.

```
> counts_data
    n11 n12 n22
1     4  53 179
2    38 104 157
3     5  50 148
4    22  77 123
5    29 102 167
6    22  76 167
7    17  98 154
8    56  96 118
```

This `history` object can be submitted to the `hm2x2prior` function, which can then return samples from the MCMC analysis of both the model parameters for the observed data and prior distributions of the new study parameters: rate-ratio $RR^*$ and $\tau^*$.

The approximate distributions of the hierarchical model priors from the MCMC samples are shown in Figure 2. Optionally, a package like `RBesT` can be used to produce a parametric approximation of the resulting prior for ease of use with the new experiment.

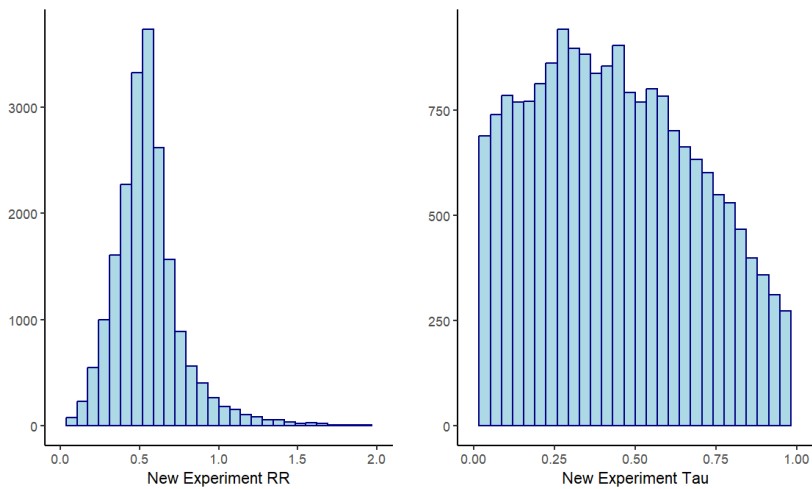

**Figure 2.** MCMC samples of hierarchical model prior distributions from 8 historic experiments.

## 6. Alternative Model

The hierarchical model described in Section 2 is convenient if interest is in the risk ratio. There are other inferential targets of interest for this model. If interest is the risk difference, the conditional probability $p_{11}/\tau$, or in homogeneity across tables, the parameterization of Johnson and May (1995) [2] might be preferred. We provide an alternative hierarchical model for these other circumstances and provide an example of how it can be used. The JAGS code for this model is found on the same github site as the earlier model.

At the first level, we assume the same data model as before,

$$z_i \sim \text{trinomial}(N_i, \mathbf{p}_i) \tag{10}$$

We reparameterize the model in terms of the marginal probability $\tau_i = p_{11i} + p_{12i}$ and the conditional probability $\alpha_i = \frac{p_{11i}}{p_{11i} + p_{12i}}$. This parameterization has the benefit that both parameters have support $[0, 1]$. The resulting likelihood is

$$L(\alpha, \tau) \propto \prod \alpha_i^{n_{11i}} (1 - \alpha_i)^{n_{12i}} \tau_i^{n_{11i} + n_{12i}} (1 - \tau_i)^{N_i - n_{11i} - n_{12i}}$$

Since both $\alpha_i$ and $\tau_i$ are defined on $[0, 1]$, we assume normal models on the logits of these parameters,

$$\phi_i = logit(\tau_i) \sim \mathcal{N}(\mu_\phi, \sigma_\phi^2) \tag{11}$$

and

$$\gamma_i = logit(\alpha_i) \sim \mathcal{N}(\mu_\gamma, \sigma_\gamma^2). \tag{12}$$

We assume the following priors for the top level of the hierarchy,

$$\mu_\phi \sim \mathcal{N}(0, 10) \tag{13}$$

$$\sigma_\phi \sim \mathcal{U}(0, 2) \tag{14}$$

$$\mu_\gamma \sim \mathcal{N}(0, 10) \tag{15}$$

$$\sigma_\gamma \sim \mathcal{U}(0, 2). \tag{16}$$

As mentioned previously, there are several options for priors on the between-trial standard deviation, with the uniform being one option along with half-normal and half-t distributions. The result of the analysis was that the strata level parameters were not sensitive to changes in the upper bound on the uniform between 1 and 10 and were similar to using a half-normal as well. Because there are only three groups, for large values of the upper bound on the uniform, there was some sensitivity on $\mu_\phi$ and $\mu_\gamma$. However, it is

well-known that $\sigma_\phi$ or $\sigma_\gamma$ exceeding two on the log-odds scale implies extreme differences in the probabilities between the strata, that is, probabilities for some groups close to 0 and others close to 1, which is unlikely in this case.

The risk difference for the $i$th study is $\delta_i = \tau_i - \alpha_i$. It might be of interest to test that the conditional and marginal probabilities are equal but still allow for heterogeneity across strata. That is, the difference in each stratum is zero; thus, $\tau_i = \alpha_i$. This simplifies the data model to the following likelihood:

$$L(\tau) \propto \prod \tau_i^{2n_{11i}}((1 - \tau_i)\tau_i)^{n_{12i}}(1 - \tau_i)^{N_i - n_{11i} - n_{12i}}$$

For this reduced model, only a hierarchical model for the $\tau_i$ is required, and it is the same as described above.

Finally, an intermediate model would be where the risk differences are equal across strata but still have baseline heterogeneity. That is, $\alpha_i = \tau_i - \delta$ where $\delta$ is the common risk difference. In this case, after replacing each $\alpha_i$ with $\tau_i - \delta$, the likelihood is

$$L(\delta, \tau) \propto \prod (\tau_i - \delta)^{n_{11i}}(1 - \tau_i + \delta)^{n_{12i}}\tau_i^{n_{1i} + n_{2i}}(1 - \tau_i)^{N_i - n_{11i} - n_{12i}}$$

This model requires a prior for $\delta$. For the multinomial probabilities above to be bounded between zero and one, we require

$$max(\tau_i) - 1 \leq \delta \leq min(\tau_i) \tag{17}$$

thus, we give $\delta$ a $beta_{[A,B]}(a, b)$ prior where $A = \max(\tau) - 1$, $B = \min(\tau_i)$, and $a$ and $b$ can be chosen to reflect expert opinion about the location of $\delta$. In the absence of information, setting $a = b = 1$ yields a uniform prior over the support.

For a specific dataset, the best-fitting model can be determined by comparing the Deviance Information Criterion numbers. As an example, we consider data found in Johnson and May (1995) [2]. The data is displayed in Table 6. In this example, patients are stratified by the severity of the disease, and an initial treatment is administered. After one week, patients who demonstrate improvement are discharged, while patients who do not show improvement are given a second phase and evaluated again one week later. The resulting data are provided in Table 5. We fit the models described above using the JAGS software using two chains with a 5000 iteration burn-in and inferences based on 25,000 iterations. Convergence was monitored via the Gelman–Rubin statistic and graphically with the history plots. No indication of a lack of convergence was noted. The DIC for the model where the marginal and conditional probabilities are equal is 69.41, the DIC for the intermediate model with equal $\delta$ across strata was found to be 66.93, while the DIC for the most general model was found to be 69.30. Thus, the DIC indicates that the model where the differences between the conditional and marginal probabilities are equal across strata is the best fitting.

**Table 6.** Data for two-phase treatment stratified by severity of disease.

| | | Phase II | |
| Phase I | No Imp | Imp | Total |
|---|---|---|---|
| **Mild** | | | |
| No Imp | 46 | 83 | **129** |
| Imp | | 176 | **176** |
| **Total** | **46** | **259** | **305** |
| **Moderate** | | | |
| No Imp | 16 | 37 | **53** |
| Imp | | 91 | **91** |
| **Total** | **16** | **128** | **141** |
| **Severe** | | | |
| No Imp | 6 | 21 | **27** |
| Imp | | 43 | **43** |
| **Total** | **6** | **64** | **70** |

## 7. Discussion

Numerous frequentist and Bayesian procedures have been proposed for estimating parameters in a correlated 2 × 2 table with a structural zero. The Bayesian hierarchical model we propose extends previous work in that it is the first Bayesian model to consider estimation in the context of multiple tables. Further, it provides tools to use this set of tables for use in future studies. These tools use the informative priors derived from historical data to assist with analysis and estimation or serve as design priors in sample size planning.

The simulation study illustrated that, as the sample sizes and number of studies increased, all parameters were well estimated, except the between-study standard deviation, which was overestimated. Though the overestimation diminishes as numbers of studies increase, it does not appear to go away. Adding an informative prior does improve estimation. However, even with this overestimation of the variance, the study-level parameters are very well estimated, so unless interest centers on the between-study standard deviation, we still recommend the relatively diffuse priors we used in the paper.

We are interested in expanding the applicability and deepening the effectiveness of this toolkit. One avenue for future research is the broadening of the model scope, i.e., bringing this approach to related but different table settings. Some examples are settings with the comparison of multiple second-stage treatments, or settings with $K$ tests/treatments rather than just two.

Another route for future research involves the incorporation of covariates into the use of the hierarchical model prior for both estimation and sample size planning. In settings where the focal parameters are related to the covariate mix of a given sub-population, the prior could adapt to the covariate mix of the new experiment population, weighting the experiments in the experiment history more heavily, which are more similar. This would result in a more effective use of the available information.

**Author Contributions:** Conceptualization, J.S.; Methodology, J.S. and W.S.; software, W.S. All authors have read and agreed to the published version of the manuscript.

**Funding:** This research received no external funding.

**Data Availability Statement:** Data are contained within the article.

**Conflicts of Interest:** The authors declare no conflicts of interest.

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
