# Peer review of "A Bayesian Hierarchical Model for 2-by-2 Tables with Structural Zeros"

_stats, doi:10.3390/stats7040068_

Round 1

Reviewer 1 Report

Comments and Suggestions for Authors

This paper proposes a Bayesian Hierarchical Model for analyzing 2-by-2 tables with structural zeros, focusing on correlated binary data. The proposed model integrates data from multiple studies and employs meta-analytic predictive (MAP) priors for informed prior distributions. 

My major concerns are about the simulation.

1. The simulations in the paper are a helpful starting point. It would be useful for the authors to conduct extensive simulation studies to investigate how well these simulations mimic real-world scenarios. I also recommend that the authors explore how their proposed model could be applied to a real-world data example, if applicable. This would greatly increase the relevance and practical value of their research.

2. While trace plots are commonly used to assess convergence, it might not be feasible to examine each trace plot for the simulation. Quantitative approaches such as the Gelman-Rubin diagnostic test could be a more suitable choice for assessing convergence.  Including a summary of the Gelman-Rubin diagnostic test statistics for the simulation studies would enhance the clarity of the analysis.

3. The authors did not include Type I error rates for the proposed model in the simulation studies. Could the authors comment on whether the Type I error rates are inflated or remain close to the $5\%$ nominal level? This would help clarify the performance of the model.

4.    The following papers are quite relevant, and they should be cited in the paper:

     1.  Yang et al.(2016), “Meta-analysis framework for exact inferences with application to the analysis of rare events”, Biometrics, 72, 1378-1386.

     2.  Fan et al.(2024), “Something Out of Nothing? The Influence of Double-Zero Studies in Meta-analysis of Adverse Events in Clinical Trials.” Statistics in Biosciences, https://doi.org/10.1007/s12561-024-09431-y

5. There are typos and grammar errors in the manuscript.

 On page 8, line 254, the phrase should read, ``If interest is the risk difference, the conditional probability...'' instead of repeating  ``the the ''.

Author Response

We thank the referee for the careful reading of the paper and these thoughtful suggestions. We have responded to each below.

Comments 1: [The simulations in the paper are a helpful starting point. It would be useful for the authors to conduct extensive simulation studies to investigate how well these simulations mimic real-world scenarios. I also recommend that the authors explore how their proposed model could be applied to a real-world data example, if applicable. This would greatly increase the relevance and practical value of their research.]

Response 1: This is a good and helpful point. The simulation study now focuses on a more realistic scenario where the sample sizes vary from study to study. We also took one of the simulated data sets and illustrated how to do the analysis in R. If there are further suggestions for other simulations, we would be glad to add them, either to the manuscript or to an appendix attached to the paper.

Comments 2:  [While trace plots are commonly used to assess convergence, it might not be feasible to examine each trace plot for the simulation. Quantitative approaches such as the Gelman-Rubin diagnostic test could be a more suitable choice for assessing convergence.  Including a summary of the Gelman-Rubin diagnostic test statistics for the simulation studies would enhance the clarity of the analysis.]

Response 2: This is an important point. It is true that there is no practical way to check graphics for every simulated data set. We have added the Gelman-Rubin statistic in our summary of the simulation study. The Gelman-Rubin statistic indicates the parameters converge in the simulated data sets.

Comments 3:  [The authors did not include Type I error rates for the proposed model in the simulation studies. Could the authors comment on whether the Type I error rates are inflated or remain close to the $5\%$ nominal level? This would help clarify the performance of the model.]

Response 3: Good point. For the simulations we considered, we computed intervals for all the data sets. The coverage of these intervals (which were typically at or above nominal) indicates that using these intervals to test hypotheses would adequately control the Type I error.

Comments 4: [The following papers are quite relevant, and they should be cited in the paper:

  1. Yang et al.(2016), “Meta-analysis framework for exact inferences with application to the analysis of rare events”, Biometrics, 72, 1378-1386.
  2. Fan et al.(2024), “Something Out of Nothing? The Influence of Double-Zero Studies in Meta-analysis of Adverse Events in Clinical Trials.” Statistics in Biosciences, https://doi.org/10.1007/s12561-024-09431-y]

Response 4: Thank you for these suggestions, we have added both of these references to the paper.

Comments 5: [There are typos and grammar errors in the manuscript.

 On page 8, line 254, the phrase should read, ``If interest is the risk difference, the conditional probability...'' instead of repeating  ``the the ''.]

Response 5. Thanks again for the careful review. We have corrected this typo and several others.

Reviewer 2 Report

Comments and Suggestions for Authors

This manuscript aims to propose a Bayesian hierarchical model with full Bayes inference for 2x2 contingency tables with structural zero. While the topic adds something to the existing literature, the manuscript does not seem to meet a minimum standard. I would recommend, at the least, a careful major review for technical correctness and a thorough language check. Below are my comments.

Introduction paragraph 1: 

- Table 1 is referred to, but the corresponding table is not named. The table named Table 1 is a different table. 

- Define the notation p_ij and n_ij clearly. Mention what are '+' and '-' in the table. You should not expect the reviewer to spend too much effort in deciphering notations. Provide enough explicit information in the right places for a smooth reading.

- The definitions of RD and RR are inconsistent with their formula.   

Page 2, line 55: "straightforward"

Equation (2) holds ignoring the constant term. You should mention it in the parentheses for exactness. Also, it has multiple typos.

Page 3

- Line 88: "We reparameterize model (1)"

- Line 89: RR includes p_12 whereas earlier in its definition p_11 is used.

- Line 90: RR_i is defined using tau_i, but RR_i is used to define the limits of tau_i. Fallacy of circularity.

- Equation (3) has wrong exponent terms.

- Line 97: I would call a N(0,10^2) prior not non-informative, rather weakly informative.

- Line 104: "outperform"

- Equation 6: the circularity issue mentioned above.

- Line 115: rho_p is given a gamma(e,f)

- Line 121: The link is invalid.

Page 4

- Line 199: augment with what? revise language.

- Line 261: In the likelihood equation, minus missing in the power of tau_i.

- Justify uniform priors in Equations 14 and 16.

- Lines 266-268: Unclear. The simplified equation does not make any sense to me!

.......

I did not go over Sections 3 and 4 as there are numerous mistakes mentioned above that need to be addressed. Thanks.

Comments on the Quality of English Language

A thorough language check is necessary.

Author Response

We first thank the referee for the incredibly careful reading of the manuscript and for these suggestions. We made the mistake of changing some notation right before submitting and left several careless inconsistencies the referee identified. We address each of the comments below. 

Comments 1: [Table 1 is referred to, but the corresponding table is not named. The table named Table 1 is a different table.]

Good catch. We have corrected this mistake and have also verified that other tables are appropriately labeled throughout.

Comments 2: [Define the notation p_ij and n_ij clearly. Mention what are '+' and '-' in the table. You should not expect the reviewer to spend too much effort in deciphering notations. Provide enough explicit information in the right places for a smooth reading.]

Thank you for pointing this out. We have added definitions of these quantities and tried to give more context to the table.

Comments 3: [The definitions of RD and RR are inconsistent with their formula.]

We have clarified the definitions of these quantities.  

Comments 4: [Page 2, line 55: "straightforward"]

We have made this change.

Comments 5: [Equation (2) holds ignoring the constant term. You should mention it in the parentheses for exactness. Also, it has multiple typos.]

Another careless mistake on our part. We have changed the equality to a proportional symbol and have corrected the typos.

Comments 6: [Line 88: "We reparameterize model (1)"]

We corrected this reference.

Comments 7: [Line 89: RR includes p_12 whereas earlier in its definition p_11 is used.]

This has been changed and is related to the next point.

Comments 8: [Line 90: RR_i is defined using tau_i, but RR_i is used to define the limits of tau_i. Fallacy of circularity.]

Thank you for requesting clarification here. By our unclear presentation connecting the new parameterization to the parameters in the original model, you are correct, there is a circularity. However, practically, that is not what we did. One could easily imagine that model (3) is the model we start with (instead of the original trinomial model) and for model 3, we do have the restriction that the probability is bounded above by 1/RR. We have attempted to clarify this.

Comments 9: [Equation (3) has wrong exponent terms.]

We have corrected this

Comments 10: [Line 97: I would call a N(0,10^2) prior not non-informative, rather weakly informative.]

We use this language now.

Comments 11: [Line 104: "outperform"]

We have made this change.

Comments 12: [Equation 6: the circularity issue mentioned above.]

As addressed previously, we consider model (3) the “starting point” and references to other parameterizations are not relevant to the fitting of this model.

Comments 13: [Line 115: rho_p is given a gamma(e,f)]

We have made this correction.

Comments 14: [Line 121: The link is invalid.]

The github page is now active.

Comments 15: [Line 199: augment with what? revise language.]

We have made it more clear that the likelihood for the new study is added to the model and becomes part of the estimation procedure in order to get the predictive distributions of the unknown parameters.

Comments 16: [Line 261: In the likelihood equation, minus missing in the power of tau_i.]

We have corrected this likelihood and made it proportional since we left off the constant terms.

Comments 17: [Justify uniform priors in Equations 14 and 16.]

This is a very important point. When discussing priors on between group standard deviation in Section 2, we mentioned a variety of options, proposed by Gelman et al. (2006). We reiterate this and discuss that the results do not seem sensitive to different choices.

Comments 18: [Lines 266-268: Unclear. The simplified equation does not make any sense to me!]

Thank you again for this suggestion, as we once again were not clear in our presentation. We have attempted to clarify. The simplified model assumes the risk difference is 0 across strata, thus each tau = alpha, so we only need one parameter per stratum. Then, the third model assumes the risk difference is not 0, but is the same across the strata, so alpha_i = tau_i + delta.  We have added explanations to the text.

Round 2

Reviewer 2 Report

Comments and Suggestions for Authors

Thanks for addressing my last comments. However, there is still inconsistency left as it seems to me (unless I totally missed the point). Anyway, I will only mention the following.

- Line 23-25: Convert n_i,j --> n_ij, p_i,j --> p_i,j etc., as denoted in Table 1.

- Table 1 is still inconsistent with the statement in LIne 24--"Without loss of generality, we assume those testing positive on the first test are not retested, thus n2,1 is the structural 0." You still did not define the notations n_ij/p_ij directly, like n_ij is the number of blablabla... Further, you did not say explicitly if + means testing positive (=improvement observed?), which would be natural to assume anyway. However, if I assume this, then the above statement implies n_11 should be a structural 0. 

- Also, I did not understand this sentence (Line 25) clearly---"If those testing negative on the first test are not retested, we simply change the column labels and proceed" Change what to what? Be more explicit.

- Overall, the first paragraph with Table 1 still needs careful revision for consistency. On that will depend notations in the following sections.

- All the Tables should have a description.

- You have used both the terms 'risk ratio' and 'rate ratio' throughout the manuscript. Use only one (I hope you mean the same thing by these two phrases).

In introduction first paragraph you wrote: "The parameter of interest is the risk ratio, the ratio of the conditional probability of no improvement in the second regimen given no improvement in the first regimen to the probability of no improvement in the first regimen."

In Simulation study first paragraph, you wrote: "The rate ratio is the ratio of the initial test negative response probability and the conditional probability of a negative response on the second test given the initial negative response."

Revise for consistency.

Comments on the Quality of English Language

Language suggestions:

between trial standard deviation --> between-trial standard deviation

between study variability --> between-study variability

data set --> dataset

I also noticed several typos. Run a typo check.

Author Response

Comments 1: [Line 23-25: Convert n_i,j --> n_ij, p_i,j --> p_i,j etc., as denoted in Table 1.]

Response 1: Thank you for this notational improvement. We have made the change.

Comments 2: [Table 1 is still inconsistent with the statement in LIne 24--"Without loss of generality, we assume those testing positive on the first test are not retested, thus n2,1 is the structural 0." You still did not define the notations n_ij/p_ij directly, like n_ij is the number of blablabla... Further, you did not say explicitly if + means testing positive (=improvement observed?), which would be natural to assume anyway. However, if I assume this, then the above statement implies n_11 should be a structural 0.]

Response 2: This is a great point and an obvious point of confusion due to the nature of these problems. In some cases, only subjects who test positive are included in the second group (for instance, when interest is in a secondary infection, so you must have the primary infection to be tested for the secondary). On the other hand, for diagnostic test studies, typically only those testing negative in the first test receive the second test.  We have added this explanation to the text.

Comments 3: [Also, I did not understand this sentence (Line 25) clearly---"If those testing negative on the first test are not retested, we simply change the column labels and proceed" Change what to what? Be more explicit.]

Response 3: This was our attempt to explain the previous point and as noted in previous response, we have added more text to explain that the +/- groups are problem dependent and so will depend on the context of the problem.

Comments 4: [Overall, the first paragraph with Table 1 still needs careful revision for consistency. On that will depend notations in the following sections.]

Response 4: We believe with the added context and explanations, the table is consistent with the explanation now.

Comments 5: [All the Tables should have a description.]

Response 5: Thank you for this suggestion. We have added captions to all tables.

Comments 6: [You have used both the terms 'risk ratio' and 'rate ratio' throughout the manuscript. Use only one (I hope you mean the same thing by these two phrases).]

Response 6: We did mean the same thing and now use risk ratio throughout.

Comments 7: [In introduction first paragraph you wrote: "The parameter of interest is the risk ratio, the ratio of the conditional probability of no improvement in the second regimen given no improvement in the first regimen to the probability of no improvement in the first regimen."

In Simulation study first paragraph, you wrote: "The rate ratio is the ratio of the initial test negative response probability and the conditional probability of a negative response on the second test given the initial negative response."

Revise for consistency.]

Response 7: The definition in the simulation was unnecessary and we have made sure the definition in the introduction matches the table and the rest of the paper.

Comments 8: [Comments on the Quality of English Language

Language suggestions:

between trial standard deviation --> between-trial standard deviation

between study variability --> between-study variability

data set --> dataset

I also noticed several typos. Run a typo check.]

Response 8: We have made all these changes and corrected several other typos.